# Determinants Influenced by COVID-19 Vaccine: Employing the Health Action Process Approach and the Belief in Conspiracy Theories

**DOI:** 10.3390/vaccines11040730

**Published:** 2023-03-25

**Authors:** Bireswar Dutta

**Affiliations:** English Taught Program in Smart Service Management, Department of Information Technology and Management, Taipei Campus, Shih Chien University, Taipei 10462, Taiwan; bdutta67@gmail.com

**Keywords:** vaccination, COVID-19, health, health action process approach, belief in conspiracy theory, hesitancy

## Abstract

Vaccination is considered a practical approach to improving individuals’ health behavior to fight against the COVID-19 pandemic. However, the currently manufactured COVID-19 vaccines can only work for a limited time. Thus, continuous vaccination intention is crucially essential. The current study explores critical factors influencing citizens’ continuous vaccination intentions for COVID-19 vaccines, based on the modified health action process approach (HAPA) model and belief in a conspiracy theory. A questionnaire survey was used to collect data from people living in Taiwan. Three hundred ninety responses were employed for the final investigation. The findings suggest that openness to experience, government communication, and pandemic knowledge significantly influence vaccination intention, but the COVID-19 threat is insignificant. Second, descriptive norms play a significant role in promoting vaccination intention. Third, a belief in conspiracy theories negatively influences vaccination intention. Fourth, vaccination behavior positively influences both perceived benefits and value co-creation. Fifth, perceived benefits positively impact value co-creation and continuous vaccination behavior. Finally, value co-creation has a significant influence on continuous vaccination behavior. The proposed model, the key contributor to the current study, confirms citizens’ continuous vaccination intentions in a three-stage procedure: motivation to volitional, volitional to behavior, and volitional to continuous vaccination intention.

## 1. Introduction

Vaccinating individuals is one of the critical public health measures to combat the COVID-19 Pandemic [1,2,3,4,5]. The creation of the vaccines has received a lot of attention globally ever since the genetic makeup of SARS-CoV-2 was made public in January 2020 [6,7,8]. Based on information about their effectiveness, safety, quality, risk management techniques, and increasing fatality cases, the World Health Organization (WHO) has certified several new COVID-19 vaccines for use in emergencies [7,8,9]. The WHO stated that, as of 1 June 2021, 102 vaccines were in clinical development, and 185 were in pre-clinical development.

Besides the availability, accessibility, and affordability of vaccines, vaccine indecision is a crucial barrier affecting individuals’ decision-making about vaccines [10,11]. Nevertheless, developing COVID-19 vaccines is being carried out in several countries. However, it is indisputable that the current vaccines have an expiration date [10] and will not protect individuals forever. After a certain period, they need to top-up the vaccine to defend themselves if no better way is found to fight against the virus [10,11].

Most publications have mainly focused on individuals’ intention to accept the COVID-19 vaccine, or why individuals have hesitated to be vaccinated until now [12,13]. Ung et al. [7] explored self-efficacy, response efficacy, response cost, and social norms as significantly influencing individuals’ intention to accept the vaccine. According to Pertwee et al. [13], trust, social uncertainty, and vaccine confidence positively relate to individuals’ intention to accept the vaccine. However, very few empirical studies have explored the factors that influence individuals’ continuous intention to be vaccinated, one of the most critical conditions for macro-level public health concerns for the government in the future [14].

To better cope with individuals’ continuous vaccination behavior, it is also significant to consider the issues of conspiracy theories related to COVID-19. As the literature has explored, the more individuals believed in COVID-19 conspiracy theories, the less likely they were ready to take up the vaccine [13]. Consequently, the current study includes belief in a conspiracy theory for better understanding and predicting individuals’ intention to take up continuous vaccines.

We consider citizens’ continuous vaccination against COVID-19 to be in three stages, motivation, intention, and conducted behavior, within the overall behavioral process. The process progresses from psychological awareness to behavioral intention, finally becoming actual behavior. Thus, we developed a theoretical research framework based on the health action process approach (HAPA) model and belief in a conspiracy theory to investigate individuals’ continuous vaccination behavior. Therefore, the primary aim of the current research is to explore the following fundamental questions:

Research Question 1 (RQ1): What critical determinants influence citizens’ vaccination intentions to fight against the COVID-19 pandemic?

Research Question 2 (RQ2): How do citizens’ vaccination intentions motivate continuous vaccination behavior concerning the COVID-19 vaccine?

We believe the current study findings contribute to building a comprehensive theoretical model that the government and healthcare providers can use to improve individuals’ continuous intention to take up the COVID-19 vaccine.

## 2. Theoretical Background

### 2.1. HAPA Model

A social cognition paradigm known as the health action process approach (HAPA) identifies the belief-based components of intentions, behavior, and the resulting developments. According to the HAPA model, adopting, starting, and maintaining healthy habits can be viewed as a process made up of phases, including motivation and desire [14]. This model refers to replacing health-compromising behaviors with health-enhancing behaviors [15]. Pre-intention, intention, and action are the three steps into which this model breaks down human health behavior [16].

Although the literature essentially validates the effects among the HAPA components that were predicted, there is significant variation in the effect sizes among studies [14,15]. However, the research suggests using additional features to close the frequently reported “gap” between intention and behavior [15,16]. Motivational and volitional steps are suggested explicitly for behavior change [14]. The COVID-19 danger, trust in receiving the vaccination, openness to experience, government communication, understanding of the pandemic, and pandemic-related knowledge are all included in the motivating phase [15]. The intention–behavior gap is closed during the volitional phase, providing structures involved in performing behavior after an intention has been formed [16]. Social pressure to overcome obstacles, also known as the descriptive norm, and support for the anti-vaccine movement, also known as conspiracy-theory support, are predictors in this phase.

### 2.2. Belief in Conspiracy Theories and COVID-19 Vaccination

Conspiracy theories explain significant events that incorporate covert schemes by ill-intentioned organizations [17]. Conspiracy theories are erroneous hypotheses when other explanations are more plausible [18]. Conspiracy theories have been proven to negatively affect people’s propensity to seek medical care, except in social and political contexts [13,19]. Concerns about the safety of infant vaccinations explain the drop in polio vaccination rates in some countries [17]. Conspiracy theories negatively have influenced attitudes toward preventative measures for the treatment of COVID-19 [18,19]. Therefore, more study is required to determine under what circumstances conspiracy theories may contain crucial emotional motivations.

## 3. Hypotheses Development

The COVID-19 pandemic’s perceived threat of COVID-19 caused a dramatic increase in emotional anxiety [19]. Villani et al. [20] found that perceived infectious disease threat significantly influences individuals’ motivation to adopt a healthcare behavior. Duan and Zhu [21] explored the perceived threat that COVID-19 could substantially decrease people’s motivation to adopt a healthcare behavior. Frazier et al. [22] stated a perceived high risk of harm motivates individuals to act to reduce their risk. Duan and Zhu [21] also advocated that perceived threat is a determining factor for motivating individuals to comply and take up vaccination.

**Hypothesis 1 (H1).** 
*COVID-19 threat (PT) significantly influences individuals’ intention to take up the COVID-19 vaccine.*


Early adopters of innovation and innovators typically exhibit more remarkable audacity, adaptability, and a desire for fresh ideas and techniques [23]. Adopter traits hint at a person’s level of creativity and openness. A fundamental aspect of personality is openness to experience (OE), which encompasses intellectual curiosity, artistic sensitivity, liberal ideals, and emotional distinctiveness [24]. People who love novelty and variety are distinguished from those who value and seek out routine and familiarity by OE. People with high OE usually display flexible behavior and thoughts.

The literature has shown that OE, creativity, and invention go hand-in-hand. People with high OE are more inclined to act creatively [24,25]. OE has also been connected to healthy behaviors and the utilization of preventative healthcare, such as influenza vaccination [23,25]. Based on the best knowledge, no studies have examined how OE affects behaviors relating to COVID-19.

**Hypothesis 2 (H2).** *Openness to experience (OE) positively influences individuals’ intention to take up the COVID-19 vaccine*.

Su et al. [26] indicated that evidence-based health communication approaches are vital for reducing vaccination hesitancy and developing a positive attitude towards vaccines. The government must consider proper communication approaches to counter misinformation exposure and anti-vaccination conspiratorial information [27], improve individuals’ awareness, and build a trustable environment to encourage individuals to take up vaccines [28], as timely and trusted responses from Government play an important role during a health crisis. According to Su et al. [26] better exchange of ideas during the COVID-19 pandemic can reduce fear and panic, and taught how to engage in appropriate prevention action.

**Hypothesis 3 (H3).** *Government communication positively influences individuals’ intention to take up the COVID-19 vaccine*.

Pandemic knowledge refers to individuals’ consciousness and the collection of information about a pandemic’s manners of transmission and prevention. A lack of proper knowledge was identified as a critical barrier in the literature, leading to a lack of intention toward COVID-19 preventive measures [29,30]. Aerts et al. [30] explored the importance of pandemic knowledge in improving help-seeking behaviors. Individuals with a higher level of knowledge about a pandemic are more likely to seek help than others [29,30]. Therefore, examining the effect of COVID-19 knowledge level on individuals’ intentions to take up the vaccination is crucial.

**Hypothesis 4 (H4).** *Pandemic knowledge positively influences individuals’ intention to take up the COVID-19 vaccine*.

The social system is the aspect of the social structure that may impact a person’s attitudes toward innovation. An important factor that explains the adoption of innovation is the belief that the behavior will be practiced by the majority of the population inside the social system, sometimes referred to as the descriptive norm [31]. One’s intention to adopt and sustain an invention may rise under the social pressure induced by a descriptive norm. More significantly, perceived immunization descriptive standards were linked to the intention to vaccinate against COVID-19 [31].

**Hypothesis 5 (H5).** *Descriptive norm (DN) positively influences individuals’ intention to take up the COVID-19 vaccine*.

A belief in conspiracy contributes to vaccine hesitation [18]. Individuals who pursue vaccination information online are susceptible to disclosing misinformation and anti-vaccine beliefs, which could lessen their vaccination intentions [17]. The commonly accepted conspiracy theories regarding vaccination intention contend that vaccinations are harmful, but this fact is covered up to maintain profits [13]. Negative assertions about vaccine efficiency generally impacted vaccine uptake. Previous studies indicate anecdotes about vaccination efficacy being intended for societal and political advantages are not recent. Such anecdotes affect vaccination take-up from time to time [16]. Because COVID-19 is a synthetic disease, immunization might cause infertility and will be used to implant microchips into people to control them [13,14]. The COVID-19 vaccine is a messenger RNA (mRNA) vaccine that might change people’s deoxyribonucleic acid (DNA), transform them into genetically modified humans, or cause diseases and infertility to reduce the world’s population; this is the anecdote that has received the most attention. According to research by Zhang et al. [14], those with access to online information about the COVID-19 vaccine are more likely to be misled and vaccine-hesitant.

**Hypothesis 6 (H6).** *Belief in conspiracy theory (BCT) negatively influences individuals’ intention to take up the COVID-19 vaccine*.

Intention is a critical factor in health behavior change and an essential predictor of clarifying and predicting behaviors. The HAPA model verifies that individuals’ intentions positively and significantly affect their behavior [15]. Regarding COVID-19 vaccination, individuals’ positive intentions translate into their vaccination behavior.

**Hypothesis 7 (H7).** *Intention to take up COVID-19 (INT) vaccine positively influences vaccination behavior*.

People’s behavioral intentions more or less reveal how they will act, and how willing they are to participate in a particular activity [32]. Positive behavioral action can evaluate the individual’s perception of co-creation benefits and value [33]. It is a method by which an individual’s eagerness to carry out the actual activity is finally measured [33,34]. The literature explores positive behavioral action to improve individuals’ perception of benefits and value co-creation [34,35,36]. We also believe that individuals’ action toward taking the COVID-19 vaccine mostly depends on their perception of the benefits and value co-creation of taking up the COVID-19 vaccine.

**Hypothesis 8 (H8).** *Vaccination behavior positively influences perceived benefits*.

**Hypothesis 9 (H9).** 
*Vaccination behavior positively influences perceived value.*


Perceived benefit is an individual’s perception of the potential that acting will bring a positive outcome. Perceived benefits may have a beneficial impact on customers’ intention to continue using services for a variety of applications, according to published research [37,38]. Wang [38] explored perceived benefits that significantly affect users’ continuous intention to use healthcare information technology and improve its perceived value.

**Hypothesis 10 (H10).** *Perceived benefits positively influenced perceived value*.

**Hypothesis 11 (H11).** 
*Perceived benefits positively influenced continuous vaccination.*


User empowerment has emerged as a significant determinant in the healthcare literature over recent decades [39,40]. It has been contended that user empowerment has played a crucial role in the shift from healthcare being regarded as a product to its perception as a service, where individuals play an essential part [36]. In other words, individuals have considered value-creation resources that contribute to competitive advantage and have a crucial role in improving and innovating the provided services.

The literature has argued that when individuals take an active role in the process and develop mastery in managing their ongoing healthcare behavior, it results in increased value and continuous intention [39,41]. Therefore, individual empowerment is positively associated with value co-creation and improvement of continuous intention [36,40].

**Hypothesis 12 (H12).** 
*Individuals’ value co-creation positively influenced continuous vaccination.*


Based on the above discussion, the following research framework was proposed (Figure 1).

## 4. Materials and Methods

### 4.1. Designing Measurements and Surveys

With the use of a variety of techniques, the current study constructed and validated the suggested model. The construction of the proposed study model included a review of the literature and in-depth discussions with subject matter specialists from the healthcare industry and academic worlds. Through focus group discussions, the responses from the empirical inquiry were further developed into a decisive conversation and interpretation. The study model was then experimentally tested by conducting a survey using the research instrument designed for the project. The origins of the items are detailed in Appendix A.

A literature review specifies a different number of items for each factor of the current study. The test of Cronbach’s alpha and factor loadings was exercised to resolve the disparity in the number of items. The value of Cronbach’s alpha is evidence of an instrument’s quality, and a 0.7 or higher factor loading shows the factor extracts sufficient variance from that variable. Finally, the Delphi technique was conducted to collect expert-based judgments about the scales and evaluate the present study’s feasibility and outcome.

The literature indicates that demographic features affect individuals’ intentions to take the vaccine [10,21]. As a result, demographic data was also included in the questionnaire. The study’s aims are described in the first portion of the questionnaire, which comprised three sections altogether. The second half of the questionnaire has multiple-choice questions regarding demographic information such as gender, age, educational background, and place of residence. Meanwhile, the third section contains questions about study constructs.

### 4.2. The Delphi Method

The Delphi method was used to collect expert opinions in the medical field to gain consensus on research items validating the initial conceptual framework proposed in the current study. There was an all-knowledge panel of experts. Two of the four members of the committee were male assistant professors. Only one female professor was included on the panel. Each panelist had over 12 years of experience as a health information scientist. A senior health informatics associate with more than 10 years of hospital experience was the other panelist. All the professors had doctoral degrees. The age of three of the four experts was 40–50, and the other was 52. Their research fields are considerably more closely tied to health information science, so inviting them to participate on a panel poses essential questions. An outline of the research project and potential volunteers for a pilot study were suggested following two rounds of expert debate. We rated the following statements on a five-point Likert scale: strongly disagree, disagree, not sure, agree, and highly agree. There were three-week gaps between each round of the expert panel talks to eliminate the memory effect. The questionnaire was modified to improve the content validity in response to advice from subject-matter experts and relevant literature reviews.

The mean value, standard deviation, and internal consistency were all assessed once the experts received the questionnaire. In the initial round, the Cronbach’s alpha values ranged from 0.762 to 0.956. The expert group suggested three changes: (1) simplification of linked and challenging questions; (2) systematic presentation of the questions; and (3) the addition of one item to the value co-creation scale. Additionally, two items were removed, since they were unnecessary. As a result, 46 options were offered for the Delphi method’s second round.

The Cronbach’s value varied from 0.868 to 0.989 for specific paths and was 0.828 for others after the second round. Each factor’s average significance range, with a standard deviation between 0.5 and 1.5, was 3.86 to 5.00. The second-round conference’s recommendations state that awareness of the pandemic rose from seven to eight and conspiracy theories from six to seven. The other factors are the same.

### 4.3. IRR Index

Rater agreement was measured using the Kappa statistic. Hair et al. [42] deem an amount (agreement) exceeding 0.6 reasonably acceptable. According to the most recent research, the agreement level of 0.718 is statistically significant at the p < 0.001 level (Table 1).

The internal consistency of the research items was verified during each iteration of the Delphi procedure using Cronbach’s alpha (α). Growing uniformity was evaluated as a measure of homogeneity for the evaluations and a potential indicator of agreement among the panelists.

On a scale from one to five, the respondents were asked to score their level of comprehension. One is the lowest level, and five is the highest. Then, for each panelist, the average value of each item is calculated. The item was accepted if the average value was three or above. Unless it was more than three, the value was ignored. Cronbach’s alpha (α) and Kappa Statistic were also evaluated to decide whether to include the item in the final analysis.

### 4.4. Data Collection

People from Taiwan made up the study’s target demographic. Modern artificial intelligence techniques are used to locate, extract, and analyze subjective data in surveys in a positive, negative, or neutral fashion [43]. These tools include natural language processing (NLP), text analytics, and data science. The convenience sampling method is employed in the current study [44] because it is economical, has been applied frequently in information system (IS) research, and enables the researcher to get preliminary data and trends without the inconveniences associated with employing a randomized sample.

Due to the continuing COVID-19 pandemic, online survey methodology has become the favorite way to collect data [45]. It is also preferable for the respondents as they can keep their identities anonymous. Previous databases were employed to convey the survey. Additionally, professional and personal interactions were contacted to collect the required number of participants for the study. Thus, we prepared the final items (Appendix A) and shared the link through social sites such as emails, LinkedIn, LINE, WhatsApp, and Facebook Messenger. Regular check-ups were carried out, and sensible reminders and messages were sent through emails and social media. The respondents were informed of their rights to withdraw from participation whenever they wished during the study.

### 4.5. Sampling Distribution

The formula proposed by Nduneseokwu et al. [46] was used to find an adequate sample size for final evaluation.
S = (Zscore)^2^ × p × (1−p)/(margin of error)^2^(1)
where S is the sample size, Zscore represents the confidence level (95% confidence level was selected), p is the standard deviation (0.5 is selected to ensure a large sample), and the margin of error is associated with the confidence interval (±5%). The sample size for infinite populations results in 385 (Zscore is 1.96 for a 95% confidence level). In step two, the sample size of the region (Taiwan) to the total population (23,900,579) to obtain the adjusted population size [46]:S_adjusted_ = (S)/1 + [(S − 1)/population](2)

The results of the adjusted sample size recommend that we need a minimum of 385 valid responses.

### 4.6. Data Investigation

The data investigation was performed on the two-phase approach suggested by Anderson and Gerbing [47]: firstly, measuring the discriminant and convergent validity of the proposed research framework, and then studying the hypotheses. Structural equation modeling (SEM) was used for the statistical investigation for two reasons. SEM is a multivariate method that authorizes the concurrent approximation of several equations [46]. Additionally, SEM performs factor and regression analysis in a single step, as SEM is used to examine a structural principle.

Our study explores people’s intentions to take up continuous vaccination to fight against the COVID-19 pandemic. To protect respondents’ privacy, “gender” and “Living area in Taiwan” were considered the nominal variables, while “age” and “educational level” were regarded as the ordinal values. The investigation was done within one month (from 15 December 2021 to 16 January 2022). We employed the following selection criteria: “Have you been vaccinated?” The respondents who had not been vaccinated were not included in the current study.

### 4.7. IRB

The current study did not receive Institutional Review Board (IRB) approval since the participants were not asked to divulge any personally identifying information, such as physical characteristics, genetic makeup, or psychological disorders. Moreover, no laboratory results were applied. The participants were given the questionnaire and told to complete it to the best of their knowledge and understanding of the advantages and disadvantages of receiving the COVID-19 vaccination. Based on what they know, participants selected one of five possibilities (strongly disagree to strongly agree). We also offered our participants a brief overview of the COVID-19 vaccine. This was conducted because it provided answers to two problems. First, to calm any participants’ fears about the COVID-19 pandemic brought on by current information and rumors, and, second, to establish the truth regarding the COVID-19 vaccine. Also, the respondents were informed of their right to decline to take the survey.

## 5. Results

### 5.1. Demographic Data

Three hundred ninety-five responses were collected. Five of them were considered unusable because of missing values. As a result, 390 valid responses were used for the final investigation. Table 2 reports the demographic information of the responders and shows the responders are different in gender, age, and education. The study sample was compared with a nationwide sample to indicate its representativeness and was found to align with Taiwan’s population [48].

### 5.2. Reliability and Convergent Validity

Cronbach’s alpha and composite reliability were used (CR) to evaluate the reliability and determine the model’s internal consistency. The results are shown in Table 3. With values ranging from 0.805 to 0.878 and 0.938 to 0.981, respectively, the Cronbach’s alpha and CR values for each construct are higher than the advised level of 0.70 [42], demonstrating sufficient reliability and consistency.

### 5.3. Convergent Validity

Bagozzi and Yi [49] provided two standards to evaluate the convergent validity of an instrument. (1) The composite reliability should be better than 0.7, and (2) the average variance extracted (AVE) of each construct should be greater than the variance originating from the measurement error of that construct. (the AVE should be greater than 0.50). According to Table 2, the AVE values ranged from 0.829 to 0.911, satisfying both convergent validity requirements, and the composite reliability for each construct ranged from 0.938 to 0.981, above the suggested value of 0.7. This, therefore, implies perfect convergent validity.

### 5.4. Discriminant Validity

Fornell and Larcker [50] state that the square root of the AVE of the construct must be greater than the estimated correlation between it and the other constructs in the model to assess discriminant validity. Each construct’s square root of the AVE was greater than the correlation values for that construct, meeting the requirements for discriminant validity (Table 4).

### 5.5. Tests of the Structural Model

Figure 2 displays the standardized path coefficients, path significances, and variance explained for every path (R^2^). The bootstrap resampling method is used for the significance tests for all the pathways. The variance (R-square scores) is as follows: intention to take up COVID-19 vaccine 0.421; vaccine behavior 0.682.

Table 5 lists the results of the tests conducted on the hypotheses. Except for the association between the COVID-19 threat and the intention to receive the COVID-19 vaccine, the data are consistent with eleven of the hypothesized associations. The tests of significance for all the paths employ the bootstrap resampling procedure.

### 5.6. The Structural Model’s Tests

Figure 2 displays each path’s variance, path significances, and standardized path coefficients (R^2^). In the motivation stage, the three factors, openness to experience (β = 0.27), government communication (β = −0.12), and pandemic-related knowledge (β = 0.18), significantly influence the intention to take up the COVID-19 vaccine (Figure 2). Therefore, H2, H3, and H4 were supported. However, the COVID-19 threat has an insignificant relationship to the intention to take up the COVID-19 vaccine. Thus, Hypothesis 1 is not supported.

In the volitional stage, the descriptive norm (β = 0.16) and belief in conspiracy theory (β = −0.52) positively related to the intention to take up the COVID-19 vaccine. Thus, H5 and H6 are supported.

Vaccination intention positively affects vaccination behavior (β = 0.27), so H7 is supported. In the behavioral stage, vaccination behavior positively influences the perceived benefits (β = 0.27) and value co-creation (β = 0.16). Thus, H8 and H9 are supported. Additionally, the perceived benefits positively impact value co-creation (β = 0.28) and continuous vaccination (β = 0.36). Thus, H10 and H11 are supported. The positive impact of value co-creation on continuous vaccination is significant (β = 0.27). Thus, H12 is supported.

## 6. Discussion and Implications

The role of the COVID-19 threat has demonstrated a positive influence on intention. Thus, Hypothesis 1 is not significant. This finding is not aligned with the interpretations in the HAPA model, in which risk perception is a pre-factor of behavioral intention [15]. We believe that one possible reason for this result is that this empirical analysis was carried out in Taiwan; which, to date, has controlled the COVID-19 pandemic better than others [4]. Thus, COVID-19 concerns were not as severe among people when the virus first broke out.

Conversely, people are concerned about the newly developed COVID-19 vaccines [2,4]. Vaccinated people from different regions still suffer and die due to COVID-19. As a result, people are concerned about the vaccine’s effectiveness and believe that vaccination may not be the safest health behavior to fight against the COVID-19 pandemic.

In the literature, openness to experience is considered a significant factor in motivating the intention to adopt an innovative health behavior and also plays a vital role in motivating the intention to take up the COVID-19 vaccine in the current study. Therefore, Hypothesis 2 is significant. This finding aligns with the HAPA model’s conclusions, in which self-efficacy is a developing factor in behavioral intention [16]. Highly open individuals are inclined to appreciate various new experiences [25]. They have a higher likelihood of being prepared to receive cutting-edge preventive measures such as the COVID-19 vaccine [24]. Roger [23] asserts that people more receptive to new information and experiences are more likely to be prepared and use the abilities necessary for bridging innovation from outside the conventional system [19]. The study findings indicate that intention towards vaccination may be more easily reformed based on scientific evidence of the vaccine effectiveness among those who are low in openness to experience.

Government communication is considered indispensable in to fight against pandemics, as timely information release is a critical step for the success of any public healthcare system, especially during a pandemic [28]. The current study findings found that government communication policy in the public healthcare sector positively influences individuals’ vaccine intention, in line with previous studies [26,27]. Thus, Hypothesis 3 is significant. Governments should announce timely vaccine information, such as vaccination strategies, modalities, and actions, to improve transparent and comprehensible public communication with public engagement [27]. An effective vaccination program needs regulatory agencies to communicate efficiently to engage citizens and confirm the highest collaboration with the suggested vaccination processes. The current study’s findings indicated that government health agencies should cooperate carefully with the different media types to develop better communication with the citizens and publicize appropriate information on COVID-19 vaccines.

Since COVID-19 poses a new threat to individuals’ health, the level of COVID-19-related knowledge significantly influences vaccination adoption intention, and our findings supported this hypothesis. This finding is the same as the result of the HAPA model, in which self-efficacy and outcome expectancy are pre-evolving behavioral intention factors [14,16]. The higher the knowledge level, the higher the actual adoption intention of the COVID-19 vaccine. From the perspective of COVID-19, when citizens perceive a positive outcome of vaccination, they are more willing to perform these behaviors. The empirical finding shows that citizens will exercise pandemic-related knowledge by assessing the positive consequence of behavior. This finding confirms previous studies indicating that pandemic knowledge is relevant in contexts of high uncertainty [29,30].

The current analysis found a positive link between the descriptive norm and the intention to take the COVID-19 vaccine. The findings show that one’s perception of other people’s thoughts significantly predicts engaging in creative health activity [31]. Since the people living in Taiwan are inclined to observe the authorities and follow social norms [32,33], others also accept that the recommendation of accepting the COVID-19 vaccination might improve their probability of performing the behavior. Considering that more people in the societal system adopt an innovation also lessens the uncertainty associated with the innovative concepts. It improves individuals’ acceptance that the behavior is relevant and endorses the intention to participate in the behavior.

The current study findings indicated that belief in conspiracy theory undermines COVID-19 vaccine intention. The result for H6 confirms that belief in conspiracy theory negatively affects individuals’ intention toward immunization. Speculation about vaccine preparation, development, circulation, and efficiency negatively influences citizens’ immunization readiness. The success of the immunization program is in jeopardy due to these adverse opinions spreading through communities and becoming widely held views.

Vaccination behavior significantly impacts the perceived benefits and perceived value, respectively, in line with the literature [35,36]. From the perspective of COVID-19, individuals compare their vaccination experience with their expectations after performing vaccination behavior. If their expectations are achieved, they will sense that vaccination brings benefits and their action brings value. The empirical findings explore that their expectations can be confirmed when citizens are vaccinated. They found that their activity eventually promised to reduce the associated risk of a pandemic. Proper communication and openness to experience innovation improve citizens’ perception of the effectiveness of the COVID-19 vaccine. So far, the number of casualties of vaccination in Taiwan, resulting in death or severe cases, is less than 1%, which is almost negligible.

## 7. Limitations and Future Study

The study’s findings are from a single inquiry with participants in Taiwan, and an online survey was used to collect data. Therefore, investigators must be cautious when applying the findings to other healthcare environments, as respondents with biases may select themselves into the sample. Thus, future studies must investigate a cross-cultural context to evaluate divergences in antecedents to practice intentions. Second, 390 valid responses are used in the final analysis. While the response rate is reasonably acceptable for individuals’ intention to take up COVID-19 vaccination, the sample size is moderately small. Comparatively, the low sample size may have set up a selection bias, and we are not convinced that the study findings represent Taiwan’s total population. Therefore, the current study’s findings could be considered a foundation for broader research to improve the external validity of those conclusions. Third, we found an intense relationship between predictor variables and endogenous constructs. Future studies should explore the relationships based on personality characteristics such as age, gender, residency, and income. As the literature has explored, personality differences significantly influence behavioral beliefs and the ability for decision-making. Thus, the findings could assist policymakers in comprehensively understanding how different personalities affect citizens’ intentions and lead to building positive motivation to take up vaccines.

## 8. Conclusions

The current study examines critical factors influencing citizens’ intentions to take up continuous vaccines to fight against the COVID-19 pandemic through an integrated model, derived from the classical theory of HAPA and conspiracy theory, with the descriptive norm, perceived benefits, and perceived value to explain how societal pressure and individuals’ consideration of the usefulness of performing an action influence their vaccination intention. The findings indicate conspiracy theories negatively influenced citizens’ vaccination intention. Even individuals who believe in conspiracy theories deter others from taking-up vaccines. The COVID-19 threat couldn’t severely influence the people living in Taiwan, as the vaccination program was implemented promptly and successfully. Additionally, the government communication policy distributed well-timed, precise, and transparent information to inspire citizens to take up the vaccine, which reduced the threat. Our findings show that citizens are optimistic about taking the vaccine more than once [51]. The significance of engaging citizens open to experience influences early diffusion development, as they could encourage others in society to be vaccinated. Citizens considering their healthcare behavior can support the fight against the pandemic and create societal value.

## Figures and Tables

**Figure 1 vaccines-11-00730-f001:**
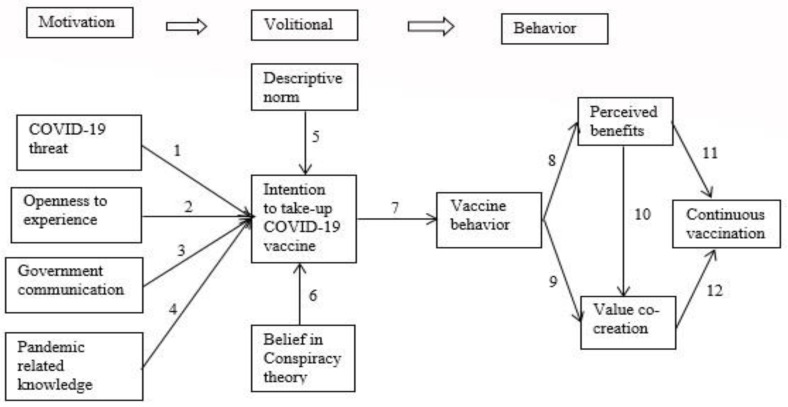
Research Framework.

**Figure 2 vaccines-11-00730-f002:**
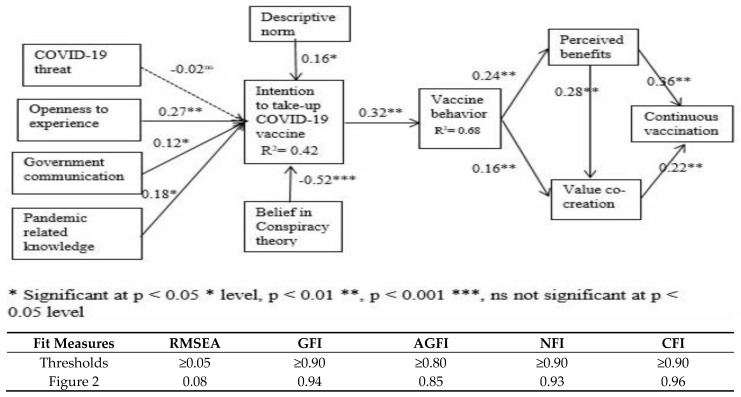
Path analysis result.

**Table 1 vaccines-11-00730-t001:** Kappa analysis.

	Value	Asymp. Std. Error ^a^	Approx. T ^b^	Approx. Sig.
Measure of Agreement Kappa	0.718	0.088	7.992	0.000
N valid for cases	390			

^a^ Not accepting the null hypothesis. ^b^ Using the asymptotic standard error, assuming the null hypothesis.

**Table 2 vaccines-11-00730-t002:** Demographic statistics of respondents.

Item	Option	Count	Percentage %
Gender	Male	200	51.39
	Female	190	48.61
Age	18–29	149	38.33
	30–39	103	26.39
	40–49	74	18.89
	50–59	40	10.28
	≥60	24	6.11
Education Level	Elementary/Junior high school	13	3.33
	Senior high school	28	7.22
	College	246	63.06
	Master’s or above	103	26.39
Living area in Taiwan	North	134	34.36
	Central	99	25.38
	West	12	3.07
	South	117	30.00
	East	28	7.19

**Table 3 vaccines-11-00730-t003:** An assessment of the measurement quality.

Construct	Cronbach’s Alpha	Composite Reliability (CR)	Average Variance Extracted
PT	0.820	0.950	0.866
OE	0.823	0.981	0.891
GC	0.869	0.976	0.829
PK	0.867	0.968	0.911
DN	0.822	0.951	0.867
BCT	0.878	0.985	0.858
INT	0.805	0.938	0.874
VB	0.847	0.974	0.865
PB	0.884	0.942	0.862
VC	0.817	0.956	0.878
CVN	0.868	0.916	0.867

**Table 4 vaccines-11-00730-t004:** Correlation of the measured constructs.

	PT	OE	GC	PK	DN	BCT	INT	VB	PB	VC	CVN
PT	**0.930**										
OE	0.679 **	**0.943**									
GC	0.521 **	−0.16 **	**0.910**								
PK	0.543 *	−0.13 **	0.478 *	**0.954**							
DN	0.446 **	−0.21 **	0.521 **	0.692 **	**0.931**						
BCT	0.451 *	−0.27 **	0.442 **	0.541 *	0.494 **	**0.926**					
INT	0.559 *	−0.25 **	−0.12 **	−0.14 **	−0.10 **	−0.18 **	**0.934**				
VB	0.479 *	−0.31 **	0.188 *	0.186 **	0.378 *	0.256 *	0.478 *	**0.930**			
PB	0.364 **	−0.33 **	0.194 **	0.126 **	0.218 *	0.267 **	0.512 *	0.468 *	**0.928**		
VC	0.261 **	−0.17 **	0.218 **	0.154 **	0.312 **	0.186 **	0.342 **	0.278 **	0.352 **	**0.937**	
CVN	0.376 *	−0.41 **	0.234 *	0.138 **	0.426 *	0.242 *	0.298 *	0.126 **	0.156 **	0.286 **	**0.931**

** correlation is significant at the 0.05 *, 0.01 ** level (two-tailed).

**Table 5 vaccines-11-00730-t005:** Result of hypotheses testing.

Hypothesis	Proposed Hypothesis Relationship	Path Coefficients	*t*-Statistics	Hypothesis Test Results
H1	PT → INT	−0.02	1.127	Rejected
H2	OE → INT	0.27	2.562	Supported
H3	GC → INT	0.12	4.934	Supported
H4	PK → INT	0.18	4.864	Supported
H5	DN → INT	0.16	4.216	Supported
H6	BCT → INT	−0.52	4.825	Supported
H7	INT → VB	0.32	2.956	Supported
H8	VB → PB	0.24	2.341	Supported
H9	VB → VC	0.16	2.162	Supported
H10	PB → VC	0.28	2.786	Supported
H11	PB → CVN	0.36	3.264	Supported
H12	PV → CVN	0.22	2.452	Supported

## Data Availability

All the data are available within the published paper, and the questionnaire used for the current study is included in the Appendix A.

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
