# Peer review of "Determinants Influenced by COVID-19 Vaccine: Employing the Health Action Process Approach and the Belief in Conspiracy Theories"

_vaccines, 2023, doi:10.3390/vaccines11040730_

Round 1

Reviewer 1 Report

The authors have tried to develop a model to predict the acceptance of repeated vaccinations in Taiwanese population. They first used Delphi method with 4 professionals to validate the research hypothesis, whereafter they performed a questionnaire with 390 persons responding to test their hypothesis. I have, however, difficulties to believe that the whole process of hypothesis generation and the testing its validity is a robust one. First of all, there were only 4 experts in the Delphi panel making it rather unreliable. Furthermore, there are totally 12 hypothesis that the authors put in a row in a rather arbitrary order. The authors should give better evidence on the validity of the hypothesis generation as well as better evidence that the questionnaire data puts the multiple hypothesis just in the order that the authors now suggest.  

Author Response

  1. The authors have tried to develop a model to predict the acceptance of repeated vaccinations in Taiwanese population. They first used Delphi method with 4 professionals to validate the research hypothesis, whereafter they performed a questionnaire with 390 persons responding to test their hypothesis. I have, however, difficulties to believe that the whole process of hypothesis generation and the testing its validity is a robust one. First of all, there were only 4 experts in the Delphi panel making it rather unreliable. Furthermore, there are totally 12 hypothesis that the authors put in a row in a rather arbitrary order. The authors should give better evidence on the validity of the hypothesis generation as well as better evidence that the questionnaire data puts the multiple hypothesis just in the order that the authors now suggest.

Ans. Thank you for your comment. I try to explain these hypotheses were formed based on the discussion of literature and in-depth discussions with specialists from the healthcare industry and academicians, which will be found in Lines 213-233.

Additionally, the IRR index was used to measure Rater agreement using the Kappa statistic to provide validation regarding the selection of items, which will be found on Lines 264-279.

With the use of a variety of techniques, the…validated the suggested model. Line 213-214

The construction of the proposed study…the healthcare industry and academic worlds. Line 214-216

Through focus group discussions, the… conversation and interpretation. Line 216-217

The study model was then experimentally…origins of the items are detailed in Table S1. Line 218-220

A literature review specifies a different number…each factor of the current study. Line 221-222

The test of Cronbach’s alpha and factor loadings…the disparity in the number of items. Line 222-223

The value of Cronbach’s alpha is… extracts sufficient variance from that variable. Line 223-225

Finally, the Delphi technique was…evaluate the present study's likelihood and outcome. Line 225-226

The literature indicated that demographic… intention to take the vaccine [10,21]. Line 227-228

As a result, demographic data was… also included in the questionnaire. Line 228-229

The study's aims were described in… which comprised three sections altogether. Line 229-230

The second half of the questionnaire has… background, and place of residence. Line 230-232

At the same time, the third section contains… about study constructs. Line 232-233

Rater agreement was measured using the Kappa statistic. Line 264

Hair et al. [47] deem an amount (agreement) exceeding 0.6 reasonably acceptable. Line 264-265

According to the most recent research, the… significant at the p<.001 level (Table 1). Line 265-266

The internal consistency of the research items… using Cronbach's alpha (α). Line 270-271

Growing uniformity was evaluated as a measure…agreement among the panelists. Line 271-273

On a scale from 1 to 5, respondents… asked to score their level of comprehension. Line 274

One is the least suitable number, and… five is the most appropriate. Line 275

Then, for each panelist, the average… value of each item is calculated. Line 275-276

The item was accepted if the Average… value was three or above. Line 276-277

Unless it is more than 3, the value is… include the item in the final analysis. Line 277-279

Reviewer 2 Report

This is of relevance study to understand the structure of determinants associated with booster administration of SARS-CoV-2 vaccination. The methodology is well-considered and described. The following issues, however, should be addressed before publication.

1) Ethical procedures; The institutional review board did not review this study due to an anonymized online survey. The author should declare that all study participants consented online before responding to the questionnaires. The description should be mentioned in the main text and in the footnote section.

2) Participants recruitment; The authors described the artificial intelligence for participant recruitment in section 4.3. The variables to represent Taiwan’s population should be mentioned. Did the proportions of variables in Table 1 represent the population?

3) Limitation; The authors indicated an intense relationship between predictor variables and endogenous constructs in lines 452 to 453. Please describe the relationship in this context in detail. Were there associations between participant demographics and constructs in the survey?

4) Conclusion; the authors said that citizens were optimistic about taking the vaccine more than once in lines 470 to 471. However, we do not know the proportion of participants who agree with continuous vaccination. This article demonstrates the association of contracts for continuous vaccination.

Author Response

This is of relevance study to understand the structure of determinants associated with booster administration of SARS-CoV-2 vaccination. The methodology is well-considered and described. The following issues, however, should be addressed before publication.

Ans. Thank you.

1) Ethical procedures; The institutional review board did not review this study due to an anonymized online survey. The author should declare that all study participants consented online before responding to the questionnaires. The description should be mentioned in the main text and in the footnote section.

Ans. Thank you for your suggestion. I include a paragraph regarding IRB approval, which will be found on Lines 327-338.

The current study did not receive… genetic makeup, or psychological disorders. Line 327-330

Moreover, no laboratory results… of receiving the COVID-19 vaccination. Line 330-332

Based on what they know; participants… (Strongly disagree to strongly agree). Line 332-333

We also offered our participants… because it pro-vided answers to two problems. Line 333-335

First, to calm any participants' fears… the truth regarding the COVID-19 vaccine. Line 335-337

Also, respondents were informed… their right to decline to take the survey. Line 337-338

2) Participants recruitment; The authors described the artificial intelligence for participant recruitment in section 4.3. The variables to represent Taiwan’s population should be mentioned. Did the proportions of variables in Table 1 represent the population?

Ans. Thank you for your comments. I include the reference to indicate how the study samples align with Taiwan’s population, which will be found on lines 344-346.

The study sample is compared with a… sample to indicate representativeness. Line 344-345

However, the study population aligns with Taiwan's population [47].  Line 345-346

3) Limitation; The authors indicated an intense relationship between predictor variables and endogenous constructs in lines 452 to 453. Please describe the relationship in this context in detail. Were there associations between participant demographics and constructs in the survey?

Ans. Thank you for your comment. I try to explain how different personalities could provide a better understanding for policymakers to framework a vaccination policy, which will be found on Lines 483-489

Third, we found an intense relationship… variables and endogenous constructs. Line 483-484

Future studies should explore… such as age, gender, residency, and income. Line 484-486

As literature explored, personality… behavioral beliefs and ability for decision-making. Line 486-487

Thus, findings could assist policymakers… positive motivation to take-up vaccine. Line 487-489

4) Conclusion; the authors said that citizens were optimistic about taking the vaccine more than once in lines 470 to 471. However, we do not know the proportion of participants who agree with continuous vaccination. This article demonstrates the association of contracts for continuous vaccination.

Ans. Thank you for your comment. I include the reference to the assertion of the finding, which will be found on lines 502-503.

Our findings show that citizens are… about taking the vaccine more than once [51]. Line 502-503
